

# Assessment of crusting effects on interrill erosion by laser scanning

Yaxian Hu[1,2], Wolfgang Fister[2], Yao He[1] and Nikolaus J. Kuhn[2]

[1] State Key Laboratory of Soil Erosion and Dryland Farming on the Loess Plateau, Institute of Soil and Water Conservation, Northwest A&F University, Yangling, Shaanxi, China

[2] Physical Geography and Environmental Change, Department of Environmental Sciences, University of Basel, Basel, Switzerland

## ABSTRACT

**Background**. Crust formation affects soil erosion by raindrop impacted flow through changing particle size and cohesion between particles on the soil surface, as well as surface microtopography. Therefore, changes in soil microtopography can, in theory, be employed as a proxy to reflect the complex and dynamic interactions between crust formation and erosion caused by raindrop-impacted flow. However, it is unclear whether minor variations of soil microtopography can actually be detected with tools mapping the crust surface, often leaving the interpretation of interrill runoff and erosion dynamics qualitative or even speculative.

**Methods**. In this study, we used a laser scanner to measure the changes of the microtopography of two soils placed under simulated rainfall in experimental flumes and crusting at different rates. The two soils were of the same texture, but under different land management, and thus organic matter content and aggregate stability. To limit the amount of scanning and data analysis in this exploratory study, two transects and four subplots on each experimental flume were scanned with a laser in one-millimeter interval before and after rainfall simulations.

**Results**. While both soils experienced a flattening, they displayed different temporal patterns of crust development and associated erosional responses. The laser scanning data also allowed to distinguish the different rates of developments of surface features for replicates with extreme erosional responses. The use of the laser data improved the understanding of crusting effects on soil erosional responses, illustrating that even limited laser scanning provides essential information for quantitatively exploring interrill erosion processes.

Corresponding author
Yaxian Hu, huyaxian@nwafu.edu.cn

# INTRODUCTION

After decades of improvement by researchers and farmers worldwide, risks of field-scale soil loss can be predicted reasonably well by erosion models such as the Revised Universal Soil Loss Equation (RUSLE) (*Wischmeier & Smith, 1978*) or the Water Erosion Prediction Project (WEPP) (*Flanagan & Nearing, 2000*). In the latter model, and several other process-based models (e.g., EUROSEM, CREAMS), the contribution of interrill erosion to the overall soil loss has been accounted for in separate sub-models (*Knisel*

& Nicks, 1980; Morgan et al., 1998; Aksoy & Kavvas, 2005). Interill erosion comprises soil loss by non-concentrated and raindrop-impacted flow (Kinnell, 2005). Fine and light particles enriched in organic matter are often entrained and transported selectively, leading to preferential removal of soil organic carbon and phosphorus, and their transfer into watercourses as non-point source pollution (Sharpley, 1985; Quinton, Catt & Hess, 2001; Lal, 2003; Teixeira & Misra, 2005; Warrington et al., 2009; Kuhn et al., 2012; Hu, Fister & Kuhn, 2013). Therefore, realistic simulation of interrill erosion is essential. A key problem for such modeling is the parameterization of soil resistance to crust formation during one or a sequence of erosion events (Kuhn, Bryan & Navar, 2003; Kuhn & Bryan, 2004; Kinnell, 2005; Hu, Fister & Kuhn, 2016).

When subjected to raindrop impact, the soil surface is compacted, and soil aggregates experience destruction through slaking, swelling, micro-cracking and dispersion (Le Bissonnais, Bruand & Jamagne, 1989; Le Bissonnais, 1996; Darboux & Le Bissonnais, 2007). The overall impacts of these changes of soil surface properties on interrill erosion are ambiguous because they can induce the formation of cohesive structural crusts, which stabilize the soil surface and protect it from further erosion (Chen et al., 1980; Le Bissonnais, 1990). However, a compacted surface is also smoother, leading to shallower and faster runoff, which increases runoff erosivity (Le Bissonnais, Renaux & Delouche, 1995; Quinton, Catt & Hess, 2001; Kuhn, 2007). Meanwhile, the destruction of aggregates generates fine, loose mineral or aggregated fragments that can be transported easily by raindrop-impacted thin flows (Kuhn, Bryan & Navar, 2003; Anderson & Kuhn, 2008). Smoothing of the surface, as well as the formation of local surface irregularities also induce differences in runoff depth and thus the way raindrop energy is dissipated on soil surface (Kinnell, 2005). Depending on the flow depth, this may protect soils from erosion, but can also enhance detachment and transport (Torri, Sfalanga & Chisci, 1987).

The effect of crust formation is not limited to the amount of erosion, but also affects the quality of sediment. Due to its limited runoff energy, erosion by non-concentrated flow often moves fine and light particles, including the substances attached to them, selectively (Basic et al., 2002; Schiettecatte et al., 2008; Kinnell, 2012). Aggregates at the soil surface continue to breakdown into smaller fragments during a rainfall event, discharging loose materials of different sizes and densities (Chen et al., 1980). Sediment of varying size compositions is thus enriched differently with nutrients and organic matter, posing unknown pollution risks to downstream watercourses (Kuhn et al., 2012; Hu, Fister & Kuhn, 2013). The variations of sediment composition do not only depend on the changes of stability and transportability of surface materials over time, but are also affected by surface roughness and its interaction with rainfall and runoff (Moore & Singer, 1990; Kuhn, Bryan & Navar, 2003). Furthermore, on soils with mixed texture and stable small aggregates, the initial removal of loose minerals and aggregated particles progressively exposes cohesive structural crust, leading to a distinct temporal pattern of initially high and then declining erosion rates (Moore & Singer, 1990; Kuhn, Bryan & Navar, 2003). In addition, as repeatedly reported in previous studies (Nearing, Govers & Norton, 1999; Armstrong et al., 2011; Hu, Fister & Kuhn, 2016), the inter-replicate variability of runoff and soil erosion rates under strictly controlled experimental conditions also illustrates a

complex, fine-scale interaction between soil surface microtopography, crust formation and soil particles, including their spatial patterns, that is hardly overcome even by adding more replicates.

The above-described examples of the interaction between crust formation and interrill erosion illustrate the need for spatial data capturing the resistance to erosion (hereafter referred to as erodibility) of soil surface and its change over time. Currently, the lack of such quantitative information introduces uncertainties for interrill erosion modeling (*Darboux & Le Bissonnais, 2007*; *Bremenfeld, Fiener & Govers, 2013*; *Hu, Fister & Kuhn, 2016*). Apart from characterizing the composition and stability of the crust, the change of soil surface roughness over time is an essential variable controlling soil resistance to interrill erosion and sediment quality (*Kuhn, Bryan & Navar, 2003*; *Armstrong et al., 2012*; *Kuhn & Armstrong, 2012*). However, soil microtopography changes are difficult to monitor (*Issa et al., 2004*; *Algayer et al., 2014*). Studies of soil micromorphology, e.g., *Chen et al. (1980)*, do not capture the appropriate scale, whilst indirect measurements of roughness on larger areas, for example by detecting directional reflectance of crusts (*Anderson & Kuhn, 2008*; *Croft, Anderson & Kuhn, 2012*), only provide an index value for roughness. Other recently developed imaging techniques, such as Structure from Motion (SfM), have also been applied to detect small soil surface variations (*Vinci et al., 2017*; *Krenz & Kuhn, 2018*) or to point out the spatial distribution of erosion and deposition hotspots at plot scale (*Remke et al., 2016*; *Krenz & Kuhn, 2018*). Terrestrial or airborne laser scanning have also been widely used to investigate slope-scale rill erosion or catchment-scale gully morphology (*Vinci et al., 2015*; *Wu et al., 2018*). However, all these studies and methods lack the millimeter-scale resolution required for the assessment of crust formation and its effects on flow hydraulics and soil resistance to erosion (*Eltner et al., 2015*). A method such as laser scanning, at least in a laboratory setting, enables the generation of the most precise digital elevation models. *Huang & Bradford (1992)* demonstrated that laser scanning at a 0.5-mm grid resolution can be used to replace speculative explanations of soil erosional responses with more quantitative interpretations. *Abban et al. (2017)* employed a laser scanner and several indices derived from the obtained data to decipher the role of rain splash on surface roughness. All these studies illustrate that laser scanning has the potential for closing the gap between the ideal quality of data required to assess crusting effects on interrill erosion and a feasible acquisition of soil microtopography in a laboratory setting.

In this study, we carried out high-resolution laser scanning on a laboratory flume to measure changes in crust microtopography before and after a rainfall event. The data were obtained during an experiment that focused on the temporal dynamics of soil erosion by raindrop-impacted flow on two soils of similar texture, but with different land management, organic matter content and aggregate stability (*Hu, Fister & Kuhn, 2013*). We hypothesized that the high-resolution laser scanning signals were sufficiently sensitive to detect minor changes of soil microtopography induced by crust formation, and thus can be employed to quantitatively explain differences of soil erosional responses between soils subject to different land management. Such use of laser scanning would avoid the often qualitative explanations for differences in interrill erosion between events or soils of different quality.

**Table 1  Selected properties of the conventionally farmed soil (CS) and organically farmed soil (OS).** Different superscripted letters in each column indicate significant differences ($t$-test). The subscripted numbers after each average value show the standard deviation ($n = 10$). For more soil properties, please refer to *Hu, Fister & Kuhn (2013)*.

|     | Clay (%) | Silt (%) | Sand (%) | Percentage of aggregates >250 μm (%) | Soil organic carbon (mg g$^{-1}$) |
|-----|----------|----------|----------|--------------------------------------|-----------------------------------|
| CS  | $16.8^a_{1.38}$ | $71.47^a_{1.76}$ | $11.50^a_{1.00}$ | $66.85^a_{0.47}$ | $10.9^a_{0.05}$ |
| OS  | $14.39^b_{0.52}$ | $75.84^b_{0.56}$ | $9.77^b_{0.38}$ | $77.76^b_{1.87}$ | $16.9^b_{0.10}$ |

## MATERIALS & METHODS

### Experimental design and soil sampling

Two silty loams of similar texture, but different soil organic carbon content (SOC) and aggregate stability (Table 1), were subject to prolonged rainfall simulations in this study. The surface of the two soils were laser scanned before and after the rainfall events to detect the crusting-induced changes of soil microtopography and the effect on erosion by raindrop-impacted flow. As part of a series of experiments, detailed information on soil properties and experimental design have been described in *Hu, Fister & Kuhn (2013)*. In brief, the two soils, one conventionally (CS) and one organically managed (OS), were sampled in 2010 from A-horizons on two farms near Möhlin (47°33′N, 7°50′E) in northwest Switzerland. After drying at 40 °C, the two soils were sieved into 1 to 8 mm to exclude over-sized clods and to allow the observation of relative changes in surface roughness during erosion processes. Immediately after sieving, each of the soils was filled into a round flume (Fig. 1A), with an outer ring of 50 cm in diameter and an opening of 10 cm in the center (Fig. 1B). The slope between the outer edge and the inner opening was 10%. To facilitate drainage, the floor of the flumes was perforated and covered by a fine cloth and a layer of sand (approximately 2 cm). This design of the flumes created a relatively uniform pattern of erosion processes along a short slope with a surface area that generated sufficient runoff and sediment for further analysis.

### Prewetting and prolonged rainfall simulations

In order to facilitate the generation of runoff and enable the observation of the effects of aggregate breakdown during crusting over the actual rainfall events on the following day, the two flumes were first subjected to a pre-wetting rainfall at an intensity of 30 mm h$^{-1}$ for 30 min right after the soil had been placed into the flume (Fig. 1C). The intensity of the actual rainfall was also 30 mm h$^{-1}$, but it lasted for 360 min, long enough to achieve prolonged steady-state runoff and associated erosional responses. A Fulljet nozzle (Spraying Systems 1/4 HH14WSQ) was used to generate multiple-sized raindrops ($D_{50}$ of 2.3 mm) with an average kinetic energy of 113.9 J m$^{-2}$ h$^{-1}$ (measured by a Joss Waldvogel-Disdrometer). While the lower kinetic energy of the rainfall compensates to a certain extent for the high intensity, the design of the experiment did not aim at the strict simulation of natural conditions, but focused on identifying the effects of crust formation on erosion and sediment properties over time.

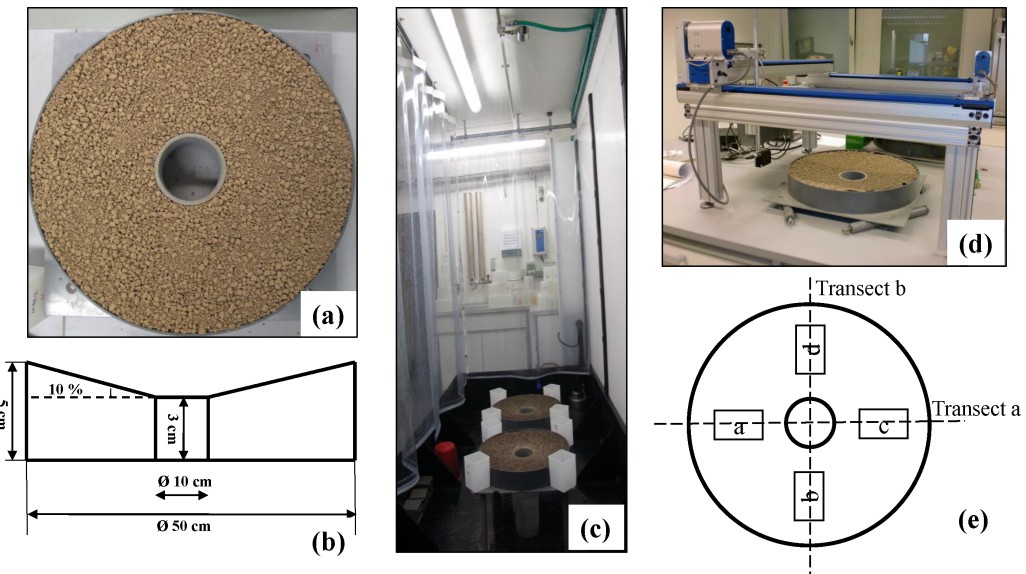

**Figure 1  Soil materials and experimental design.** (A) Layout of round flumes; (B) cross-section profile of the round flume; (C) Set-up of rainfall simulation experiment; (D) set-up of laser scanner; (E) scanning routes of two transects and four sub-plots.

During the 360 min rainfall events, runoff and sediment samples were collected every 30 min from the inner ring in the center. The total weight of the sediment suspension and the dry weight of the sediment particles were measured to calculate runoff and soil erosion rates. The SOC concentration of each sediment sample was determined using a LECO RC 612 at 550 °C. The enrichment ratio of eroded SOC (ERsoc) was calculated between the SOC concentration of the eroded sediment and that of the original soil (*Hu, Fister & Kuhn, 2013*). After the 360 min prolonged rainfall events, the round flumes were oven-dried at 40 °C, and the loose material on each round flume was then collected by a vacuum cleaner and weighed (after scanning). The entire experiment was repeated twelve times, but due to technical failure, only the data from the latter 10 replicates (replicate number 3 to 12) were used in this study.

## Laser scanning of soil surface

To capture the changes of soil surface induced by rainfall and erosion, each flume was laser-scanned three times (Fig. 1D): first in the dry condition after filling the soil in the flume (hereafter termed as "Before"); second, after the 30 min prewetting event (hereafter termed as "Prewetted"); and finally, after the 360 min rainfall event (hereafter termed as "After"). The laser was connected to a laptop and data recording was carried out using a MatLab routine. The scanned data were recorded as $x$-, $y$-, and $z$-values.

During each laser scanning, two transects (a, b) and four sub-plots (a, b, c, d, each 5 cm × 18 cm) were scanned (Fig. 1E). The two transects crossed at the center of the flume, and each transect was scanned stepwise at a 1 mm resolution, in sequence detecting 500 scanning points (Fig. 1E). On each of the four subplots (each 90 cm²), the 500

scanning points were visited automatically following the path calculated by the scanning system, and in total covered about 19% of the entire eroding area (1,884 cm$^2$) (Fig. 1E). Since similar raindrop-impacted flow would be dominant on the erosion plots, the soil surface microtopography was presumed to be spatially uniform on each plot. Therefore, two transects and four subplots were considered adequately representative to investigate erosion-induced changes of surface microtopography. In order to ensure precise alignment, the flume was accurately positioned at bottom left-hand corner of the scanning frame prior to each scanning run (Fig. 1D) and accordingly coordinated as (0, 0). By doing so, we made sure that each time the same area of the flume was scanned.

The laser scanner used in this study (Fig. 1D) was custom-designed and built at the University of Basel based on a design used by *Anderson & Kuhn (2008)* and *Croft, Anderson & Kuhn (2009)*. It consisted of a combined laboratory laser and sensor, manufactured by Baumer Electric, Frauenfeld, Switzerland (model number OADM 2014471/S14C), working at a wavelength of 675 nm and suited to measure distances of up to five meters at an accuracy of 0.1 mm (*Brunton, 2004*). The laser was mounted in one-by-one metre frame where it could be moved with stepper motors in a pre-programmed way using a Stepper Motor Controller CSD 315 (Isel Automation, Germany).

## Calculation of soil surface elevation across two transects and four subplots

Since the laser scanning was conducted stepwise over two transects and four subplots with strict alignment, the linear route of the former and the programmed route of the latter enabled a pairwise comparison of the elevation of individual scanning points. In addition, although interrill erosion did not form concentrated flow on the small round flume used in this study, the tilted soil surface provided a predominant runoff direction, which was parallel to the scanned transects. Therefore, with limited scanning data and pairwise comparison, the height distribution of scanning points was considered adequate to serve the purpose of comparing the minor variations in surface microtopography induced by crusting and their impacts on soil erosional responses (*Croft, Anderson & Kuhn, 2009*; *Vinci et al., 2015*).

By conducting regression analysis with the laser signals (the space between the scanner frame and the soil surface), the actual height of each point was calculated using Eq. (1):

$$H_i = max\{z_1, z_2, z_3, \ldots, z_{500}\} - (z_i \times 4.8619 + 0.1491) \tag{1}$$

Where, $H_i$ is the height of each data point (mm); $z_i$ is the distance between the laser and soil surface of each data point; $max\{z_1, z_2, z_3, \ldots, z_{500}\}$ is the longest distance between the laser and soil surface of all the points (namely the zero-level elevation); $i$ is in sequence from the 1st to the 500th scanning point; the constants 4.8619 and 0.1491 are the regression coefficients.

To exclude the distorted points around the opening in the center, as well these immediately nearby the outer ring (Fig. 1B), only two subsections on each side of the two transects were analyzed: $91 \leq i \leq 200$ on the left half and $311 \leq i \leq 420$ on the right half of the $X$ axis. Furthermore, to eliminate the bias introduced by the original slope

steepness and to better reflect the relative surface height changes at local scale, the heights of each data point along the two transects were standardized by the slope steepness and the distance from the lowest edge to the targeted point (Eqs. (2) and (3)). The sample protocol was also applied for the subsections of the upper and lower half of the $Y$ axis.

$$h_i = H_i - (\max\{x_{91}, x_{92}, x_{93}, \ldots, x_{200}\} - x_i) \times 10\% \qquad (2)$$

or,

$$h_i = H_i - (x_i - \min\{x_{311}, x_{312}, x_{313}, \ldots, x_{420}\}) \times 10\% \qquad (3)$$

As there were no center opening or edge effects on the four subplots (Fig. 1E), all the 500 data points of each subplot were analyzed. Since the bias possibly introduced by the original slope steepness of 10% was systematic and limited to the four subplots with small areas (5 cm × 18 cm), the $H_i$ of each scanning point inside the subplots was not standardized to slope steepness in this study. Moreover, to quantitatively compare height distributions in the four subplots, all the measured heights were then classified into eight height classes: <3 mm, 3–4 mm, 4–5 mm, 5–6 mm, 6–7 mm, 7–8 mm, 8–9 mm and >9 mm. To visualize the changes of surface roughness after erosion events, the variogram analysis of the four subplots were conducted using GS+ (Geostatistics for the Environmental Sciences). Kriging regression was applied to give the best linear unbiased prediction of the intermediate values, which were then employed to plot a 2-D version of soil surface height distribution for each subplot. In addition, the height differences between Before, Prewetted and After tests with the least and most eroded replicates were also compared to detect whether the erosion processes were the same, but just operating at different rates, or whether the soil surfaces developed in different ways and thus leading to different erosion processes.

## RESULTS

### Soil erosional responses and enrichment ratio of eroded SOC

The soil erosional responses observed during the experiments are reported in detail in *Hu, Fister & Kuhn (2013)* and only summarized here briefly (Table 2). Rates of runoff and soil erosion, and enrichment ratio of organic carbon in the eroded sediment (ERsoc) showed clear temporal patterns. The runoff on the CS increased and reached a steady state of 12.9 mm h$^{-1}$ state after 180 min, while the runoff on the OS required 240 min to stabilize at 10.7 mm h$^{-1}$. As runoff increased over rainfall time, the soil erosion rate of the CS increased first, peaked when runoff rate reached steady state and decreased afterwards. The ERsoc of the CS peaked at 1.94 around 150 min, while that of the OS reached only 1.44 after 330 min (Table 2). Further information on soil erosional response on the CS and OS are listed in Table 2.

A further noteworthy result is the inter-replicate variability, which remained between 15 and 39% even after the maximum runoff and erosion were reached (*Hu, Fister & Kuhn, 2016*). Out of the ten times repeated simulations, the least and most eroded replicate for the CS were CS-4 and CS-11, and replicate OS-9 and OS-12 for the OS (Table 3). Typically, the total runoff, soil erosion and SOC loss of the most eroded replicates nearly doubled

**Table 2    Selected soil erosional responses on the conventionally farmed soil (CS) and organically farmed soil (OS) over the 360 min simulated rainfall events.** The subscripted numbers after each average value show the standard deviation ($n = 10$).

|  | CS | OS |
|---|---|---|
| Time to initiate runoff (min) | 60 | 120 |
| Time to reach runoff steady-state (min) | 180 | 240 |
| Time to reach peak ERsoc (min) | 150 | 330 |
| Average runoff rate at steady-state (mm h$^{-1}$) | $12.9_{\pm 0.2}$ | $10.7_{\pm 0.2}$ |
| Average peak ERsoc | $1.92_{\pm 0.12}$ | $1.44_{\pm 0.05}$ |
| Total runoff on average (mm) | $55.6_{\pm 9.1}$ | $34.1_{\pm 6.0}$ |
| Total soil loss on average (g) | $27.4_{\pm 5.0}$ | $16.1_{\pm 3.0}$ |
| Total SOC loss on average (mg) | $369.1_{\pm 85.1}$ | $326.0_{\pm 59.1}$ |
| Loose materials remained on dried flumes (g m$^{-2}$) | $10.96_{\pm 3.01}$ | $43.78_{\pm 11.40}$ |

**Table 3    The erosional responses of the least and most eroded replicates on the conventionally farmed soil (CS) and organically farmed soil (OS) over the 360 min simulated rainfall events.**

| | | Total rainfall (mm) | Total runoff (mm) | Total soil loss (g) | ERsoc peak | Total SOC loss (mg) |
|---|---|---|---|---|---|---|
| Least eroded replicate | CS-4 | 184.55 | 39.00 | 15.53 | 2.01 | 239.16 |
| | OS-9 | 182.20 | 22.55 | 12.20 | 1.38 | 239.70 |
| Most eroded replicate | CS-11 | 181.18 | 66.49 | 37.33 | 1.72 | 453.65 |
| | OS-12 | 188.37 | 42.95 | 22.37 | 1.46 | 441.84 |

that on the least eroded replicate, even though they received comparable rainfall amount (Table 3).

## Changes of soil surface elevations across the two transects

Figure 2 shows the changes of the soil surface at different conditions (Before, Prewetted, After and Post-dried). After 360 min prolonged rainfall, the CS surface was visibly smoother with extended flat areas and few loose material (10.96 ± 3.01 g m$^{-2}$ as listed in Table 3), whereas the OS surface was covered to a greater extent by degraded aggregates (43.78 ± 11.40 g m$^{-2}$ as listed in Table 3). The surface elevation changes of the CS and OS are illustrated in Fig. 3. For both soils, the soil surface was lowered after the two rainfall events, with a reduction most evident after the 360 min prolonged rainfall (Fig. 3).

Apart from the average changes of surface elevation, Fig. 4 further compares the height differences between the least and most eroded replicate under the three conditions (Before, Prewetted and After). The height differences between the Prewetted and Before were more closely clustered than that between the After and Before (Figs. 4A vs. 4B, 4C vs. 4D). Specifically, for the height differences between After and Before on the CS (Fig. 4B), the most eroded replicate CS-11 had more negative height differences (point clouds <0) than increases (point clouds >0) when compared to the least eroded replicate CS-4. Similar, but more frequent negative height differences, were observed on the most eroded replicate OS-12 than that of the least eroded replicate OS-9. Moreover, the point clouds

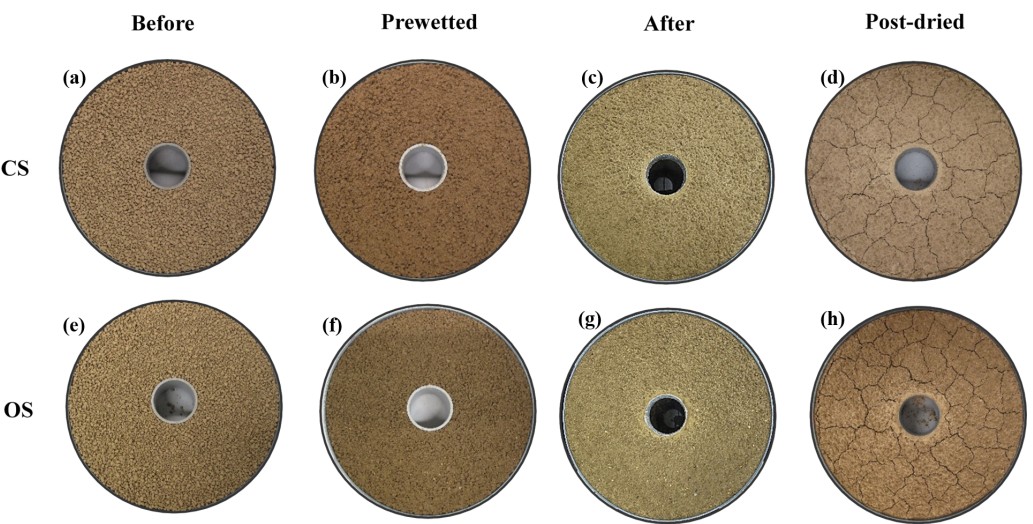

**Figure 2** Surface changes of the conventionally farmed soil (CS) and organically farmed soil (OS). (A–D) represent the CS; (E–H) represent the OS. The dark patches are degraded crumbs and blunted coarse aggregates. Light-colored areas are depositional crusts consisting of loose materials detached by raindrop impact.

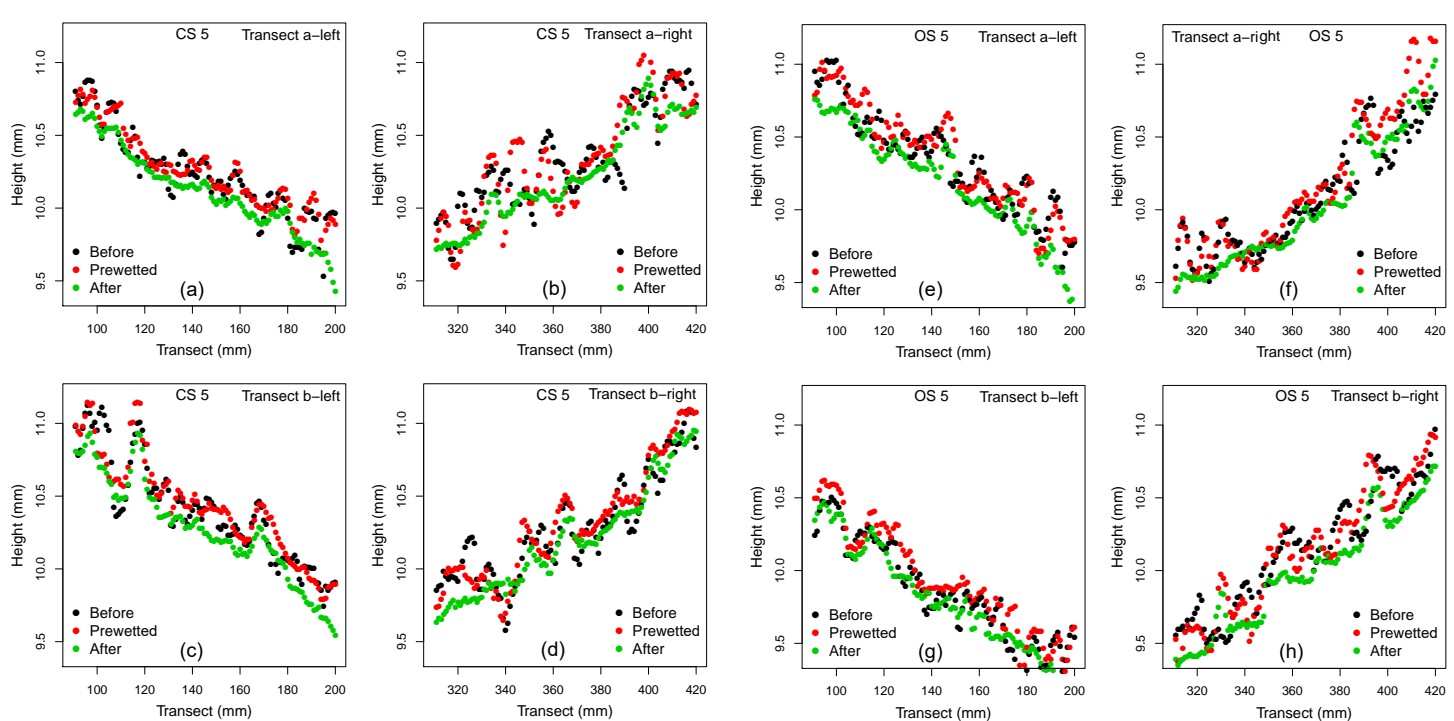

**Figure 3** Height of the individual points of the two transects (a, b) of the conventionally farmed soil (CS) and organically farmed soil (OS) Before, Prewetted and After. (A–D) represent the CS; (E–H) represent the OS. Both soils take replicate 5 as an example.

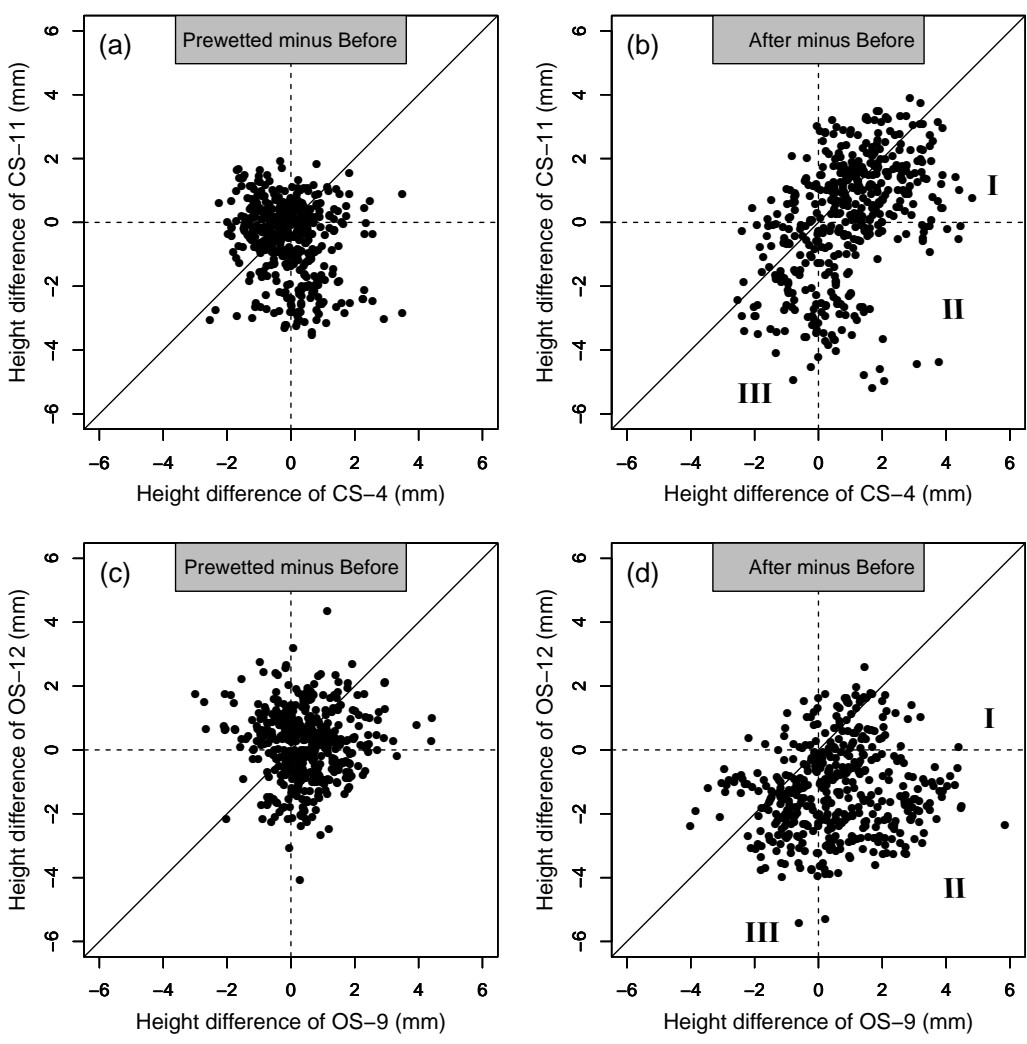

**Figure 4   Pair-wise comparison of the standardized height of the two transects on CS-4 vs. CS-11, and OS-9 vs. OS-12.** (A–B) represent the CS; (C–D) represent the OS. The replicate CS-4 and OS-9 generated the least soil loss out of the ten replicates, whereas the replicate CS-11 and OS-12 produced the most (more information in Table 3). The diagonal line in each subfigure represents the 1:1 ratio.

were also more concentrated under the 1:1 ratio for both the pair-comparison of CS-4 against CS-11, and that of OS-9 against OS-12 (Figs. 4B and 4D).

## Changes of soil surface elevations on the four subplots

Figure 5 shows that the height distribution of all the four subplots on the CS was greater and more variable than that on the OS. While the surface of both soils progressively approached the lower height classes over time (Fig. 5), this transition was much more skewed on the CS, especially after the 360 min prolonged rain (Fig. 5D). The differences of surface elevation between the two soils are also illustrated by the 2D classification in Fig. 6. The surface elevation of the two soils was quite similar when the soils were dry before the rainfall. After the 30 min of prewetting, the surface of the OS still showed a strong contrast of high and

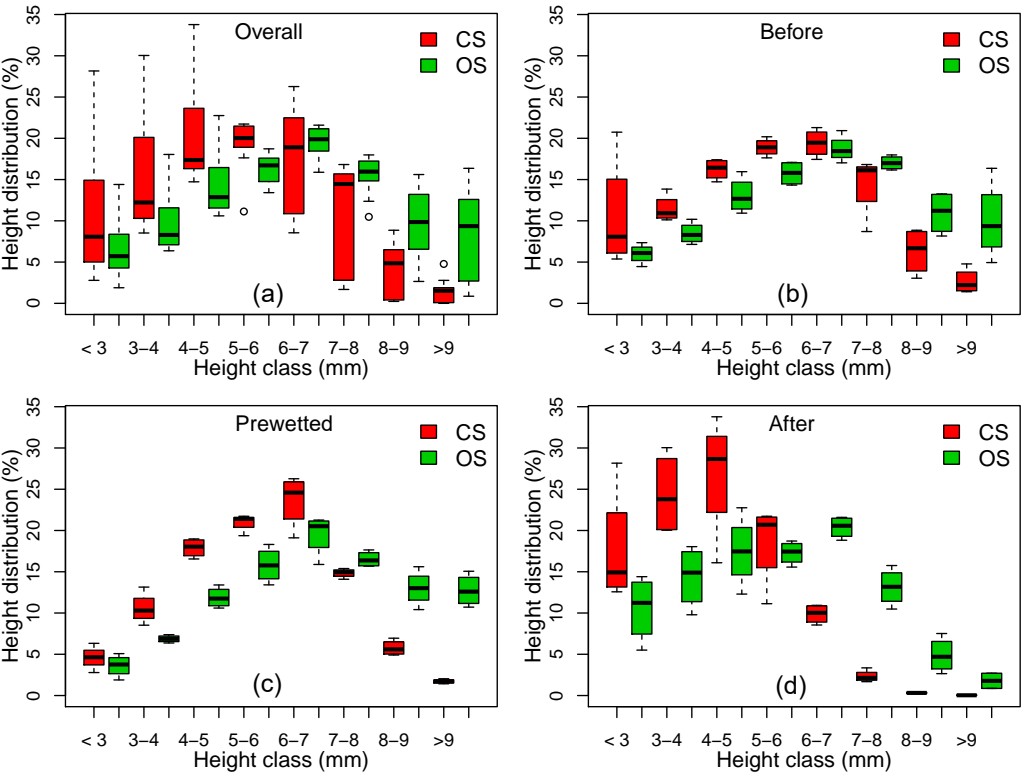

**Figure 5** **Height distribution on the four subplots of the conventionally farmed soil (CS) and organically farmed soil (OS) Before, Prewetted and After the prolonged rainfall events.** (A), (B), (C) and (D) were combined over ten replicates, namely $n = 120$ for the Overall (A), $n = 40$ for Before (B), Prewetted (C) and After (D).

low elevation (Figs. 6D and 6E), whilst the surface of the CS was noticeably flattened (Figs. 6A and 6B). After the 360 min prolonged rain, the surface height of the CS became lowered to less than 4 mm (Fig. 6C), that of the OS remained rougher between 4.5- and 6.5-mm (Fig. 6F).

## DISCUSSION

### Temporal patterns of crusting and erosion on differently structured silty loams

Given the limited runoff depth and erosion capacity of thin flow, interrill erosion is mostly attributed to raindrops impacting the flow (*Kinnell, 2005*). *Hu, Fister & Kuhn (2013)* speculated that the differences in erosion and sediment properties observed between the two soils tested in this study were associated with crust formation. They hypothesized that the different erosional responses of the similarly textured CS and OS (Table 2) reflect the influence of aggregate stability (Table 1) on surface crusting and in turn, the capacity of raindrop impacted flow for erosion. In particular, testing the hypotheses of *Hu, Fister & Kuhn (2013)* requires a quantitative assessment of the potential effects of crusting on soil surface microtopographic changes over time. The noticeably more pronounced flattening

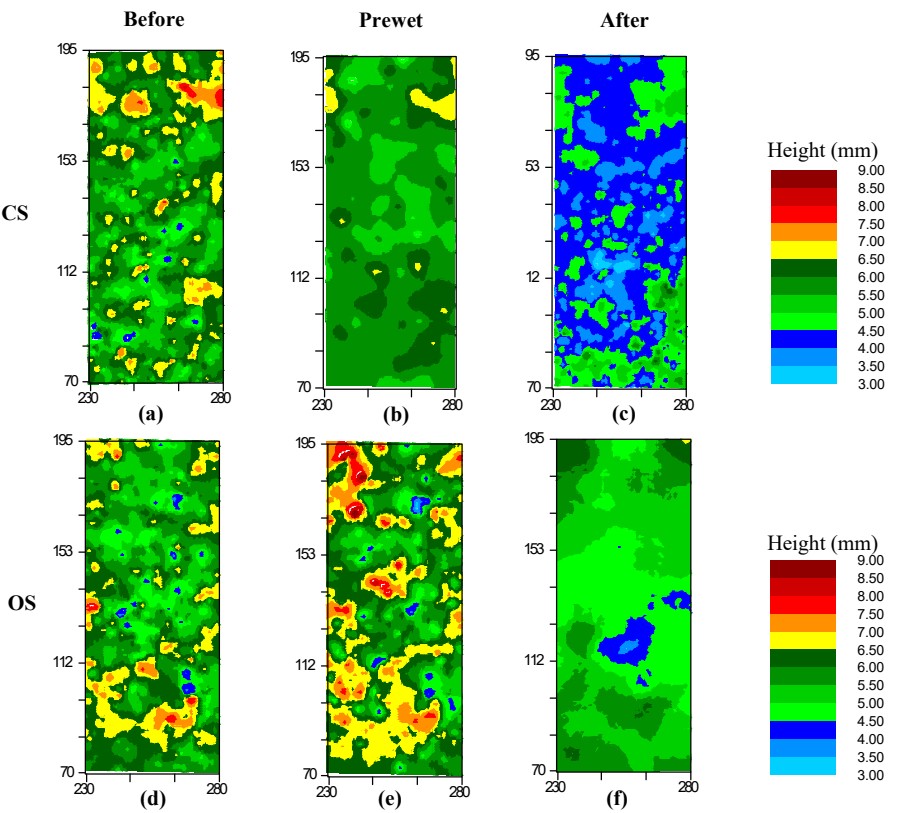

**Figure 6  Surface roughness changes of the conventionally farmed soil (CS) and organically farmed soil (OS) Before, Prewetted and After the prolonged rainfall events.** (A), (B) and (C) represent the CS, while (D), (E) and (F) represent the OS. Each was combined from the Subplot b over the ten replicates.

on the CS and the delayed deformation on the OS (Figs. 2, 3 and 5) confirm the explanation brought forward by *Hu, Fister & Kuhn (2013)*: greater aggregate stability of the OS slowed aggregate breakdown, maintaining roughness and sediment size for longer, thus also resisting raindrop impact for longer than on the CS. Such potential effects of aggregate stability on surface deformation are reflected by the more pronounced skewing toward the smaller height classes on the CS surface than on the OS surface (Fig. 5). Consequently, after the prolonged 360 min rainfall, the CS surface height was noticeably flattened to be less than 4 mm (Figs. 5D and 6C), whereas the soil surface of the OS was much rougher between 4.5 mm and 6.5 mm (Fig. 6F) and still interspersed by more loose material (Figs. 2G, 2H, Table 3). The declining soil erosion rates on the CS after its runoff rate exceeded 12.9 mm h$^{-1}$ (Table 2) indicate that runoff had overcome transport limitation and reached a supply-limited process after fine, light and loose particles had been selectively eroded (flattened surface in Figs. 5D and 6C). Judging from the abundant loose materials remaining on the OS plots (Figs. 2G, 2H, Table 2, as well in *Hu, Fister & Kuhn (2013)*, it would also eventually reach a supply-limited condition as runoff grew more competent over time by removing loose particles and exposing cohesive crust. This deduction is also

supported by the delayed decline of the ERsoc on the OS once runoff rates had stabilized (Table 2).

## Variability of crust formation and erosion identified by laser scanning

Apart from detecting the different erosional responses between the two soil types, the laser scanning also effectively captured the variations of surface microtopography among replicates. The greater amount of points with height differences <0 mm for the most eroded replicate CS-11 and OS-12 (Fig. 4D) clearly illustrates more advanced smoothening than that for the least replicates CS-4 and OS-9 (Fig. 4B). The unbalanced distributions of height differences in the pairwise comparisons (Fig. 4) further demonstrate the divergent influences of surface microtopography on the soil erosional responses of the least and most eroded replicates. To be specific, the more concentrated point clouds under the 1:1 ratio line in Figs. 4B and 4D practically represent three scenarios. (1) The positive height differences under the 1:1 ratio line in section I of Fig. 4B indicate that the scanned surface was rougher after 360 min of rainfall (After) than before, and such roughening was more pronounced in the least eroded replicate CS-4 than the most eroded replicate CS-11. (2) The concentrated cloud in section II of Fig. 4B displays that certain parts of the least eroded replicate CS-4 became rougher after the prolonged rainfall than Before (positive height differences from 0 mm to 6 mm), whereas some sections of the most eroded replicate CS-11 were flattened after the prolonged rainfall events (negative height differences from −6 mm to 0 mm). A similar, but even more obvious concentration of the point cloud can be observed in section II of Fig. 4D, illustrating the divergent development of soil surface elevation between the least and most eroded replicates of the OS. (3) The negative height differences in section III under the 1:1 ratio line of Fig. 4B suggest that those areas were smoother after the 360 min rainfall, and the most eroded CS-11 was more smoothened than the least eroded CS-4. All the three scenarios of more pronounced flattening (Figs. 4B and 4D) on the most eroded replicates CS-11 and OS-12 are consistent with their nearly doubled runoff and soil loss as opposed to the least eroded replicates CS-4 and OS-9 (Table 3).

The covariance between the observed erosional response and the laser data illustrates the effectiveness of millimeter-resolution laser scanning to detect the minor topographic changes. On the one hand, this not only confirms the decisive role of soil properties such as organic matter content and aggregate stability in crust formation, soil erosional responses and sediment properties (Tables 1 and 2) (*Hu, Fister & Kuhn, 2013*). On the other, the results clearly illustrate that laser data also help to uncover the causes of the 15%–39% inter-replicate variability among the ten simulated rainfall events (*Hu, Fister & Kuhn, 2016*) by effectively distinguishing the rates of surface feature development between the most and least eroded replicates (Fig. 4, Table 3). Therefore, with limited laser scanning data and pairwise comparison of height distributions, our observations corroborate previous studies where the changes of soil surface microtopography during rainfall events were effectively quantified by laser scanning (*Huang & Bradford, 1992*; *Abban et al., 2017*). Furthermore, our findings show that with laser scanning an improved quantitative interpretation of interrill erosion experiments is possible, which would otherwise have been attributed just

in a qualitative way to inherent variability under controlled laboratory conditions (*Bryan & Luk, 1981*; *Anderson & Kuhn, 2008*; *Hu, Fister & Kuhn, 2016*).

## CONCLUSIONS

The changes of microtopography of the surface of two soils with different rainfall-erosion interaction were measured by fine-scale laser scanning before and after the application of simulated rainfall. The surface of both soils experienced a flattening, but they displayed persistently different temporal patterns of crust development and associated erosional responses. By effectively distinguishing the minor variations of surface microtopography and thus the rates of surface feature development among the least and most eroded replicates, the height differences detected by laser scanning revealed the causes of the 15%–39% inter-replicate variability among the ten simulated rainfall events. This improved the understanding of crusting effects on soil erosional responses and demonstrated that laser scanning can be used to examine interrill erosion with more quantitative interpretations. While a promising tool in the lab, laser data of the degree of accuracy achieved in this study are likely not readily acquirable under field conditions, especially if plots are large and natural rainfalls determine crust formation (*Nearing, 1998*; *Armstrong et al., 2011*). However, studies using high resolution laser scanning conducted in the laboratory could be used to develop soil microtopography parameters that can be linked to reflectance data or DEMs derived by Structure from Motion in both laboratory and field and their relationship to interrill erosion (*Anderson & Kuhn, 2008*; *Croft, Anderson & Kuhn, 2009*). With the currently rapid development of easily accessible and affordable devices such as smart phones, high-resolution cameras and unmanned aerial vehicles, the associated digital photogrammetry and high-resolution digital elevation models (*Eltner et al., 2015*; *Vinci et al., 2015*; *Vinci et al., 2017*), there is a great potential for the development of sensible interrill microtopography parameters and the acquisition of data in laboratory and field to improve soil erosion models.

## ACKNOWLEDGEMENTS

The contributions of Ruth Strunk in carrying out the laboratory experiments are thankfully recognized. The critical discussion with Dr. Peter I. A. Kinnell greatly helped to improve the experiment design. The manuscript was also substantially improved after the proofreading by Florence Greenwood, whose passing was too premature and who is still missed by all the co-authors.

### Funding

This work was supported by the Natural Science Foundation of China (No. 41701318), the National Key Research and Development Project of China (2018YFC0507001), Innovative Research Program of the Ministry of Education, China (A315021608), and the University of Basel. The funders had no role in study design, data collection and analysis, decision to publish, or preparation of the manuscript.

## Grant Disclosures

The following grant information was disclosed by the authors:

Natural Science Foundation of China: 41701318.

National Key Research and Development Project of China: 2018YFC0507001.

Innovative Research Program of the Ministry of Education, China: A315021608.

University of Basel.

## Competing Interests

The authors declare there are no competing interests.

## Author Contributions

- Yaxian Hu performed the experiments, analyzed the data, prepared figures and/or tables, and approved the final draft.
- Wolfgang Fister analyzed the data, authored or reviewed drafts of the paper, and approved the final draft.
- Yao He analyzed the data, prepared figures and/or tables, and approved the final draft.
- Nikolaus J. Kuhn conceived and designed the experiments, authored or reviewed drafts of the paper, and approved the final draft.

## Field Study Permissions

The following information was supplied relating to field study approvals (i.e., approving body and any reference numbers):

This study not require a field permit as Severin Kym from Bäumlihof and Edi Hilpert from Eulenhof gave us their personal permission to access the field and accompanied us to collect soil samples.

## Data Availability

The following information was supplied regarding data availability: The raw data is available in the Supplemental Files.

## Supplemental Information

Supplemental information for this article can be found online at http://dx.doi.org/10.7717/peerj.8487#supplemental-information.

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
