# Peer review of "Assessment of crusting effects on interrill erosion by laser scanning"

_PeerJ, doi:10.7717/peerj.8487_

## Round 0.1 · original submission · Major Revisions

Dear authors,

I've received the reports on your manuscript from two reviewers. Both of them provided insightful reviews which I believe will help to further improve the manuscript quality.

In particular, I agree with the reviewers that the scientific objective of the research should be better clarified. Moreover, the issue raised by reviewer #2 on "surface roughness" calculation method is of utmost importance and needs to be clarified.

I am looking forward with interest to receiving your revision.

Kind regards

Reviewer 1 ·

Basic reporting

The manuscript ID 42555 describes an interesting research focused on the dynamics of crust formation (and its effects on soil erosion) in agricultural soils during a rainfall event. This research combines rainfall simulations and measurements made by using a high accurate lasing scanner. The study was carried out in laboratory with soils from two farms with different land management in Switzerland. I understand this kind of controlled experiments shall be done in laboratory but I prefer data from real field experiment because soil transport can bias the experiment. Anyway, this latter is just a personal opinion not a critical of this research I have evaluated.

The article is quite well-written and structured, i.e. it is very easy to read and understand. Since I am not a native speaker I do not feel qualified to assess the language level of the manuscript but I think its English of high quality. I have detected a mistake in the line 28. I think you have written sol (soil in French) instead of soil. Anyway, I think the most appropriated concept to be written is land management instead of soil (please see line 28) or tillage management (please see line 106).

This research is also well-motivated. The authors remain clear in the first paragraph of the Introduction section what the research gap they are trying to cover is: “to improve the parameterization (for spatio-temporal modelling) of soil resistance dynamics as a consequence of rainfall events”. Actually, you are trying to improve the temporal (pre vs. post) and spatial (reducing the cell size of DEM) accuracy of the measurements of the variable (crust) roughness (mentioned in the last paragraph of the Introduction section).

Subsequently, they explain pretty well the existing interrelationships between crust formation and soil erosion (amount and type of sediments) in the paragraphs 2 and 3 of the Introduction section. Their affirmations are well-supported by relevant literature. The references look to be well-quoted and listed at the end of the manuscript according to the guidelines for authors of the journal. In case I am wrong I apologize because each journal requires different guidelines and sometimes it is easy to be wrong in this matter.

In the paragraph 4 they justify properly the necessity of using laser scanning in order to obtain the most precise digital elevation models. Nevertheless, I have some doubts regarding the usefulness of these precise DEMs because of up-scaling. That is very good to work at the slope scale but I guess you have to make precise DEMs in many slopes in order to prepare precise inputs for any models reliable for regional scales (personal opinion). Obviously, that is not the aim of your research because you are only testing the effectiveness of laser scanning to improve the parameterization of soil resistance dynamics (crust roughness). But, in any case, reviewing an article is the best way to discuss with researchers of your same field study.

Line 106: carbon erosion? Please check this affirmation.

Figures and tables are relevant, of high quality and well-labeled and described.

The dataset provided as raw data are 144 txt files that can be perfectly managed by MO Excel/Calc or in GIS software, amongst others.

Experimental design

This research is original and useful. This information could be used by many modelers in future tools. Obviously, this research is based on a single study case (only two farms in Switzerland) but it covers many aspects. So, it is fair to say this article fits perfectly with the scope of the journal.

Regarding research questions and/or hypothesis they are not clearly mentioned, at least in the Introduction section. The authors identify perfectly the research gap and they explain properly how this gap shall be filled but there are not research questions and hypothesis to be tested addressed explicitly in the text.

The Material and methods (M&M) section is divided into 5 sub-sections that follow a logical sequence from a methodological point of view. Each methodological step is perfectly explained, i.e. this methodology could be replicable by everyone. This methodology has been made technical and ethical rigorously made (including a sufficient number of replicates). I have decided to treat each step separately in order to make easier the understanding of my comments.

Soil sampling and preparation: Experimental design and soil sampling are well-described in the previous publication (attached by the authors as supplemental material) but the line 115 should be rewritten because it is confusing. Please take into account that experimental design is the first step and soil sampling is the second one. How much time has it lasted between the finish of soil sieving and the beginning of the soil pre-wetting?

Rainfall simulations: Why are you interested in the determination of SOC eroded?

Calculation of soil surface roughness: Is it really necessary to use one sub-section for each type of measurements: 2 transects and 4 subplots?

Validity of the findings

My opinion, as reviewer, after having spent many hours in their evaluation is to “certify” the validity of these results. The process has been perfectly controlled and the results are robust. Nevertheless, I think we can discuss here about the impact and novelty of this research.

Regarding impact these findings are very useful for modelers. About novelty it depends on the personal opinion as each reader/scientist. I think it is a continuation of previous research works (kindly attached by the authors) and then it should be considered an advance in this topic.

Discussion and conclusions are consistent and they have reached my expectations as scientist. I think they are well-fitted to that the journal is grosso modo also expecting.

Additional comments

Line 122: rim or ring?

Line 195: normalized (abnormal distribution to normal distribution) or standardized (rescale the data to a known range)?

Line 253: frequent

Line 267: The fact of checking conventional farming soils flatten under low intensity rainfall events is another proof of the necessity of keeping “security” cover in our soils to keep their durability.

Line 298: crust formation (not rust formation)

Reviewer 2 ·

Basic reporting

In the paper, the laser scanning technique (coupled with a photogrammetric survey) was used to assess the crust formation. Sorry, but I don’t understand the aim of the paper, i.e. the authors would quantify the changes of the soil microtopography due to erosive processes or test the laser scanning methodology? In the first case, I think that the paper could be interesting, but the authors have to rewrite the paper in this sense. The capability of the laser scanner technology was assessed by several authors (such as Abban et al., 2017; Zheng et al., 2014; Huang and Bradford, 1992) and the novel aspect of the paper is not clear.
The literature references are poor, I suggest to enlarge the introduction and some paper that could be interesting for the paper.

Abban, B. K. B., Papanicolaou, A. N., Giannopoulos, C. P., Dermisis D.C., Wacha, K. M., Wilson, C. G., Elhakeem M., 2017. Quantifying the changes of soil surface microroughness due to rainfall-induced erosion on a smooth surface. Nonlinear Processes Geophys., 24, 569-579.
Zheng, Z. C., He, S. Q., Wu, F. Q., 2014. Changes of soil surface roughness under water erosion process, Hydrol. Processes. 28, 3919–3929.
Huang, C., Bradford, J.M., 1992. Applications of a laser scanner to quantify soil microtopography. Soil Sci. Soc. Am. J. 56, 14-21.

Experimental design

The description of the soil sampling, the rainfall simulator, and the laser scanning survey are clear.
As commented above, from Line 183 - 219, the description is ambiguous. The authors used coefficients that there are not explained in the text (eq.1).
Furthermore, the authors treated the changes of the surface roughness as the soil surface height distribution.
In the Introduction (Line 108) "close-up photography was used..." but in the section Materials and Methods there is not a description of the photogrammetric survey.

Validity of the findings

In the paper, there are some conceptual mistakes. First of all, from Line 183, the authors describe the method used to calculate the soil surface roughness. Conceptually, the soil surface roughness is very different from the actual height of the points. In literature, numerous indices and definitions of surface roughness could be found (from Kuipers, 1957 or Allmaras et al., 1966 to more recent works where the laser scanning was used such as Abban et al., 2017). However, the surface roughness is commonly quantified as the variations (using a standard deviation for example) of the surface heights in relation to a plane (Currence and Lovely, 1970) or a surface or a residual topography (Cavalli et al., 2008). If the authors evaluated the surface roughness, probably it is necessary to clarify.
Instead, if the authors want to refer the results about the differences the term "surface roughness" has to be deleted in the paper.


Kuipers, H., 1957. A relief meter for soil cultivation studies. Netherlands Journal of Agricultural Science, 5, 255-262.
Allmaras, R. R., Burwell, R. E., Larson, W. E., Holt, R. F., 1966. Total porosity and random roughness of the interrow zone as influenced by tillage. USDA Conservation Research Report 7.
Currence, H. D., Lovely, W. G., 1970. The analysis of soil surface roughness. Transactions of the ASAE 13(6).710-714.
Cavalli, M., Tarolli, P., Marchi, L., Dalla Fontana, G., 2008. The effectiveness of airborne lidar data in the recognition of channel bed morphology. Catena, 73, 249–260.

Additional comments

Probably I don't understand the novel aspect of the paper in this form.
The paper could be interesting if the authors focused on the crust formation due to erosional processes and not on the laser scanning method.
Detailed suggestions are reported on the annotated PDF file.

Annotated reviews are not available for download in order to protect the identity of reviewers who chose to remain anonymous.

---

## Round 0.2 · accepted · Accept

You have fully addressed all the issues raised by me and the two reviewers and the manuscript quality has been greatly improved after the first revision round. This is a nice piece of work, congratulations.

Reviewer 1 ·

Basic reporting

My comments and those given by the other reviewer(s) have been helpful to improve considerably this manuscript. The autos have made a good job.

Experimental design

The research rationale is perfectly explained. In addition, the methodology is both consistent and useful.

Validity of the findings

The results are very interesting. In the future I would like to replicate this methodology in my study areas.

Additional comments

Dear Authors,
I am satisfied with the changes made for the current version. So, I am going to suggest the article can be accepted in the current form.
Sincerely,
Reviewer #1

Reviewer 2 ·

Basic reporting

In this revised form the paper is clear and unambiguous.
All the critical points of the original paper were clarified.
The literature references are enlarged and the novelty and the results of the study were focused.
In this form I think that the paper could be published because the manuscript has been significantly improved.

Experimental design

No comment

Validity of the findings

No comment

Additional comments

I suggest to correct the L245 of the manuscript without revision.